# Coupling quantum corrals to form artificial molecules

Saoirsé E. Freeney, Samuel T. P. Borman, Jacob W. Harteveld and Ingmar Swart⋆

Debye Institute for Nanomaterials Science, Utrecht University, the Netherlands

⋆ I.Swart@uu.nl

## Abstract

Quantum corrals can be considered artificial atoms. By coupling many quantum corrals together, artificial matter can be created at will. The atomic scale precision with which the quantum corrals can be made grants the ability to tune parameters that are difficult to control in real materials, such as the symmetry of the states that couple, the on-site energy of these states, the hopping strength and the magnitude of the orbital overlap. Here, we systematically investigate the accessible parameter space for the CO on Cu(111) platform by constructing (coupled) quantum corrals of different sizes and shapes. By changing the configuration of the CO molecules that constitute the barrier between two quantum corrals, the hopping integral can be tuned between 0 and -0.3 eV for s- and p-like states, respectively. Incorporation of orbital overlap is essential to account for the experimental observations. Our results aid the design of future artificial lattices.



# 1 Introduction

The scanning tunneling microscope makes it possible to position adsorbates and vacancies on surfaces with atomic scale accuracy [1]. This approach has been used to explore the limits of data storage [2–5], to perform logic operations [6–9], to study chemical reactions at the single molecule level [10–13], and to study the electronic and magnetic structure of atomically well-defined structures [14, 15].

With respect to studying electronic properties of extended systems, two complementary approaches have been used. The first approach is based on coupling localized states of either adatoms, vacancies or dangling bonds, [16–23]. By positioning such species with atomic scale precision, artificial electronic molecules or lattices can be created and their electronic structure studied. Initial experiments focused on the evolution of the electronic structure with system size. However, more complicated and interesting phenomena can also be studied, such as topological states of matter and Majorana bound states [21, 24].

The second approach, following the ideas underpinning the quantum corral, is based on patterning a 2D electron gas (2DEG) with a (periodic) scattering potential. In particular, the CO on Cu(111) platform has been used to study the electronic structure of periodic and non-periodic systems [25–31]. Here, the CO molecules act as repulsive scattering centers for the surface state electrons of Cu(111) [32]. By placing these scattering centers with atomic scale accuracy, a large variety of potential energy landscapes can be created for electrons. For example, by creating a triangular lattice of CO molecules, the electrons are confined to the anti-lattice, i.e. a honeycomb geometry [26]. Density of states measurements revealed the emergence of a Dirac cone in the 2DEG, as observed in graphene. Building on this approach, an electronic Lieb-lattice [27], quasi-crystal [28] and electronic fractal [29] have been realized. Recently, it was shown that this material platform can also be used to study topological states of matter [31, 33].

One of the advantages of using artificial lattices is that it allows control over parameters that cannot be controlled easily in real materials. These include the on-site energy, the strength of the hopping parameter, orbital overlap, and which orbitals couple. However, the values for the hopping parameter, on-site energy of each electronic site and overlap are not immediately obvious given a certain configuration of CO/Cu(111). Currently, determining these parameters is an involved iterative "reverse engineering" procedure which includes first designing the lattice and performing a muffin tin calculation to check that the features of interest are observable, which may take several iterations of design changes. The resulting muffin tin band structure is compared to the output of a tight binding calculation. The tight binding parameters are then adjusted such that the tight-binding band structure matches the muffin tin result [33].

In this work, we systematically investigate the accessible tight binding parameter range for the CO/Cu(111) platform by coupling quantum corrals into artificial molecules. The report is arranged as follows. First, a background on the subject is given, and the experimental details are discussed. We show how changing the size of rectangular and triangular corrals affects their on-site energy and we determine the effective mass of the confined electrons. We specifically focus on rectangular and triangular corrals, as these allow for space-filling artificial lattices. Furthermore, we report experiments on coupling such units into dimers and trimers and extract the tight binding parameters. We investigated the coupling of both $s$-like and $p$-like orbitals. The coupling strength is adjusted with different methods; both by changing the size of the potential barrier between the corrals, and by changing the size of the corrals themselves. Finally, we studied the coupling of orbitals with different symmetries.

Before describing our results, we discuss the similarities and differences between quantum corrals and real atoms. Artificial lattices built using CO/Cu(111) can be thought of as systems

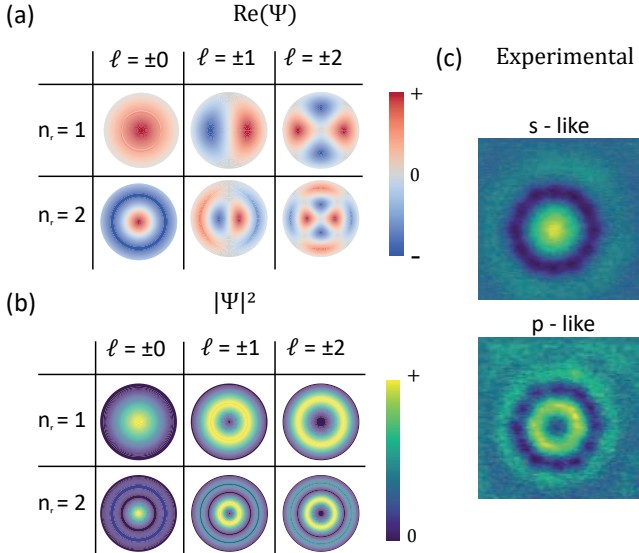

Figure 1: Modeling a circular quantum corral with the particle-in-a-box model. (a) The real part of the wavefunction enclosed in a circular well, showing its shape for different quantum numbers. (b) $|\Psi|^2$, which is proportional to the differential conductance in STM. (c) Differential conductance maps of a small quantum corral at two energies; -0.17 V and 0.21 V. These correspond to the $\ell = 0$, $n = 1$ (1$s$) and $\ell = 1$, $n = 1$ (1$p$) states.

of coupled quantum corrals. The first quantum corral was created in 1993, by positioning Fe atoms in a (nearly) circular ring on Cu(111) [1]. The electronic behavior within the corral can be readily understood in terms of a particle-in-a-box model [1,34]. Fig. 1(a) shows wavefunctions of a particle-in-a-circular-box for a combination of the first few quantum numbers. For circular corrals, the wavefunctions are characterized by the principle and angular quantum numbers, $n$, $\ell$, respectively. $n - 1$ defines the number of nodal lines in the radial direction from the center, while $\ell$ defines how many nodes occur angularly. For non-circular symmetric corrals, the angular momentum quantum number is not well-defined. However, the wave functions of circular, rectangular and triangular corrals exhibit alternation of sign and nodal line patterns that are reminiscent of nodal planes in atomic orbitals [30]. The lowest energy state has no nodal lines, the second lowest has one, etc. [35, 36]. Based on these similarities, we refer to these states of the quantum corral as $s$-like and $p$-like, respectively. The nodal line pattern of a particular state of the quantum corral can be visualized by mapping the differential conductance at the energy corresponding to that state. In principle, the spin quantum number $m_s$ is also common between a 2D particle-in-a-box and a real atom, because $m_s$ only describes whether an electron has spin $+\frac{1}{2}$ or $-\frac{1}{2}$, and is a general property of electrons.

In contrast to 2D quantum corrals, three quantum numbers appear for real atoms. The magnetic quantum number is not present in 2D systems. However, as we show below, $p_x$- and $p_y$-like states do emerge in rectangular corrals [30]. Furthermore, the allowed values of the quantum numbers are different for quantum corrals and real atoms. For example, circular 2D quantum corrals feature 1$p$-type states (see Fig 1a,b), whereas in real atoms a 1$p$ state does not exist.

In addition to Fe atoms, a variety of other adsorbates can be used as scattering centers. Because of the ease and reliability with which they can be manipulated, CO molecules are often used [6, 37, 38]. Carbon monoxide molecules on the Cu(111) surface are imaged as depressions with standard metallic tips [39]. A DFT study has suggested that this is due to

destructive interference of the protruding orbital of oxygen atom with the states in the tip [40].

Throughout this document, we show the designs of various corrals and indicate copper atoms as orange dots and CO molecules as black dots with shading that represents the apparent size of the CO molecule as viewed in STM. Corral dimensions are reported in terms of the Cu(111) lattice constant $a = 0.2556$ nm [41].

## 1.1 Tight binding description of dimers and trimers

To create artificial dimers, we construct two connected corrals with an opening between them to accommodate coupling. Fig. 2(a) shows an example of a structure consisting of two coupled rectangular corrals.

The tight binding parameters of interest are the on-site energy, $\epsilon$, the nearest and next-nearest neighbor hopping parameters, $t_1$ and $t_2$ (not present for dimers) respectively, and the overlap integral, $s$ [42,43], see Fig. 2. It was previously reported that the next-nearest-neighbor hopping integral can be non-negligible in artificial lattices [27,29–31,33]. To determine the magnitude of $t_2$, we also constructed and characterized trimers, see Fig. 2(c).

A tight binding calculation of a dimer, taking into account only the lowest energy state of each corral, results in the following expressions for the two states of the dimer

$$E_+ = \frac{\epsilon_1 + t_1}{1 + s}, \tag{1}$$

$$E_- = \frac{\epsilon_1 - t_1}{1 - s}, \tag{2}$$

where the subscript indicates the sign with which the states of the corral are added. The values of $E_+$ and $E_-$ can be directly extracted from differential conductance spectra acquired at suitable positions above the dimer (taking the shape and extent of the wavefunction into account). Since the spatial confinement of the electrons in the dimer is different from those of isolated corrals (there is an extra available area when the barrier between two corrals is

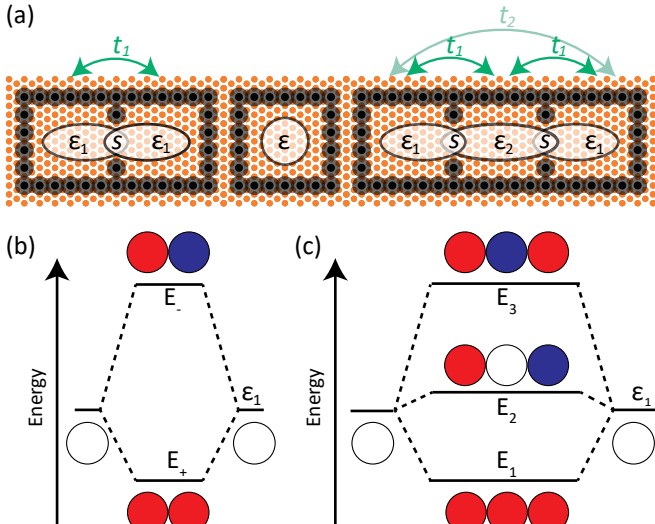

Figure 2: Coupling quantum corrals. (a) Example placements of CO molecules (black) on Cu(111) (orange) to produce a dimer, a lone corral and a trimer. White ovals roughly represent the spatial extent of the wave functions of the individual quantum corrals. (b) and (c) show the molecular orbital diagrams for a dimer and a trimer, respectively. Red represents a positive value of the wavefunction and blue negative.

removed), the on-site energy is different for coupled and individual corrals. The resulting set of two equations with three unknowns (1 and 2) cannot be solved. To determine values of $\epsilon_1$, $t_1$ and $s$, we include calculations and measurements on a trimer, as represented in Fig. 2(c). We make the assumption that the overlap integral is the same for the dimer and trimer.

In the case of a trimer, there are three energy states that correspond to bonding, non-bonding and antibonding orbitals in molecules, as illustrated in Fig. 2(e). The energies of these three states are given by equations 3, 4 and 5, respectively.

$$E_1 = \frac{\epsilon_1 + \epsilon_2 - 4st_1 + t_2 - \sqrt{(-\epsilon_1 - \epsilon_2 + 4st_1 - t_2)^2 - 4(1 - 2s^2)(\epsilon_1 \epsilon_2 - 2t_1^2 + \epsilon_2 t_2)}}{2(1 - 2s^2)}, \quad (3)$$

$$E_2 = \epsilon_1 - t_2, \quad (4)$$

$$E_3 = \frac{\epsilon_1 + \epsilon_2 - 4st_1 + t_2 + \sqrt{(-\epsilon_1 - \epsilon_2 + 4st_1 - t_2)^2 - 4(1 - 2s^2)(\epsilon_1 \epsilon_2 - 2t^2 + \epsilon_2 t_2)}}{2(1 - 2s^2)}, \quad (5)$$

where $t_2$ is the next-nearest-neighbor hopping parameter, $\epsilon_1$ is the on-site energy of each of the outer two atoms (the same as in the dimer) and $\epsilon_2$ is the on-site energy of the central atom, see Fig. 2(c). Since $E_1$, $E_2$ and $E_3$ are also observable in experiment, we now have a system of five equations (1 to 5) and five unknowns. This allows us to obtain all tight binding parameters $\epsilon_1$, $\epsilon_2$, $s$, $t_1$, and $t_2$.

## 2 Methods

All experiments were performed at $T \approx 4.5$ K in ultra-high vacuum with a ScientaOmicron LT-STM. A Cu(111) surface was prepared by several repetitions of sputtering with Ar$^+$ and annealing at 550° C. Carbon monoxide was leaked into the microscope chamber with a direct line of sight onto the Cu(111) crystal mounted in the microscope head to achieve a suitable coverage. Manipulation of carbon monoxide molecules was performed in feedback with a bias voltage of 20 mA and a current setpoint of approximately 50 nA, depending on the configuration of the tip apex. STM images were acquired in constant current mode. Differential conductance spectra and maps were acquired with the tip at constant height and using a standard lock-in amplifier technique. The frequency and amplitude of the applied modulation was 271 Hz and 10 mV r.m.s. respectively. Integration time for signal acquisition was 50 ms during spectra and 20 ms during maps. All differential conductance spectra shown have been averaged over several measurements acquired at the same position, and divided by an average of several spectra taken on bare Cu(111) with the same tip apex to minimise the LDOS contribution from the tip [26]. In each spectrum shown, the faded points represent the data after the aforementioned procedure, while the solid line represents the moving average of the same data.

Muffin tin calculations were performed to corroborate and supplement the experimental data. This technique is well-established, and has been used before to simulate results on the CO/Cu(111) platform with reasonable accuracy [27, 29–31, 33].

# 3 Results

## 3.1 Individual Corrals

We first characterize rectangular corrals. Note that because of the triangular symmetry of the underlying substrate, it is not possible to build perfectly square corrals. Fig. 3a shows the schematic structure and $\frac{dI}{dV}$ spectra of a rectangular corral with size $8\sqrt{3}a \times 14a$. Spectra taken at different positions exhibit peaks at different positions, corresponding to specific eigenstates. For example, the lowest energy level (approximately -0.3 V) has the highest local density of states (LDOS) in the center of the corral (black), whereas the next highest energy level (-0.1 V) is mainly observed away from center (at red, blue and green sites). The differential conductance maps reveal the spatial extent of these states, see the top row in Fig. 3b. The corresponding simulated maps are shown in the bottom row of the same figure. In the case of degenerate levels, the modulus squared of the relevant eigenfunctions were summed. The simulations are in excellent agreement with the experimental observations.

For rectangular quantum corrals, there are two quantum numbers that determine the energy of the system and the shape of the wavefunction; $n_x$ and $n_y$. By comparing the experimental data to the results of the particle-in-a-box model, we can assign wave functions to the differential conductance maps and peaks in differential conductance spectra. For the first few energy levels, we may draw an analogy to real atoms based on the number of nodal lines in

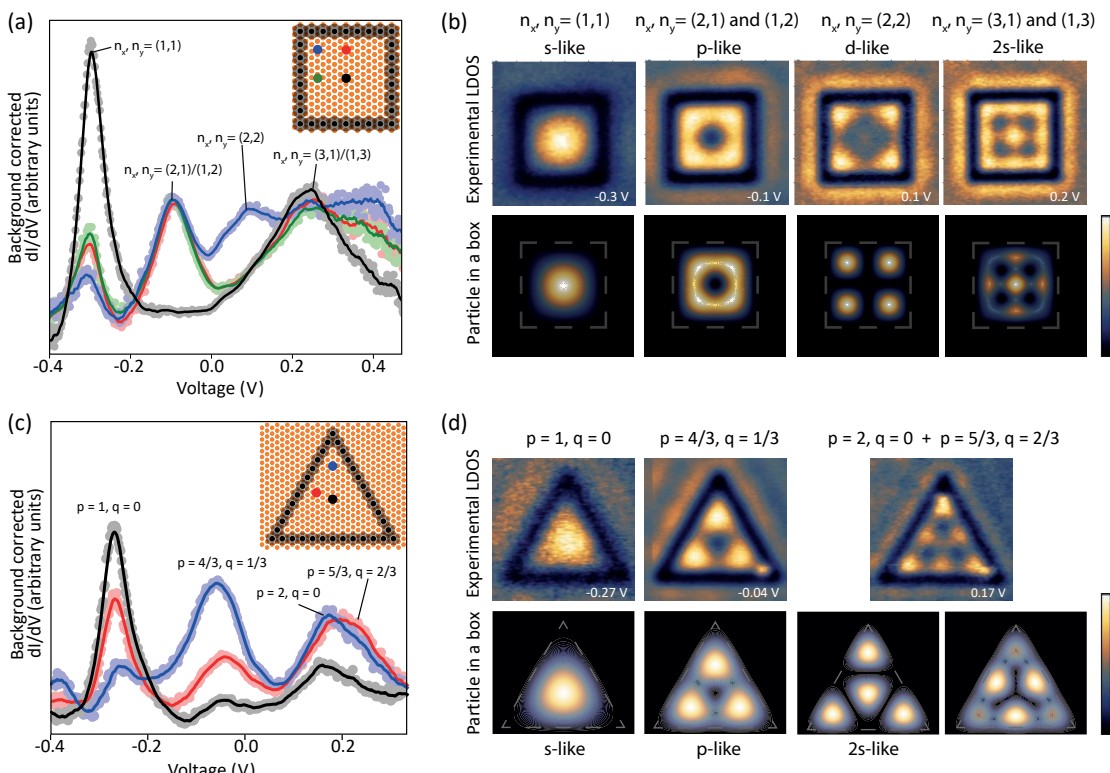

Figure 3: Differential conductance measurements on rectangular and triangular quantum corrals. (a) dI/dV spectroscopy acquired at the positions marked in the inset figure. (b) Top: experimental differential conductance maps (of size 6 nm × 6 nm) acquired at the energies stated; bottom: $|\Psi|^2$ calculated according to the particle in a box model. The quantum numbers are labeled above each peak in the dI/dV and above each LDOS map. (c), (d) same as (a) and (b) but for triangular quantum corral. The images depict an area of 6.25 nm × 6.25 nm.

$\Psi$ that intersect the center of the corral. The $n_x = 1, n_y = 1$ (no nodal lines) resembles an atomic $s$-orbital. Similarly, the $n_x = 1, n_y = 2$ (and $n_x = 2, n_y = 1$) (one nodal line) and $n_x = 2, n_y = 2$ (two nodal lines), have a similar nodal line structure as $p$- and $d-$type orbitals in atoms. The next highest state is the $2s$-like state.

We now apply the same procedure to triangular corrals. An equilateral triangular corral is constructed with side lengths $12\sqrt{3}a$. $\frac{dI}{dV}$ spectroscopy was conducted at different positions, see Fig. 3c. The lowest energy peak is observed at approximately -0.27 eV and has the highest amplitude in the center of the corral ($s-$like). state. The second energy level is mainly localized near the corners ($p$-like orbital). $\frac{dI}{dV}$ maps acquired at the peaks observed in the $\frac{dI}{dV}$ spectra are shown in Fig. 3d. For a particle-in-a-triangular-box, there are two quantum numbers; $p$ and $q$. The calculated eigenfunctions corresponding to the first four energy levels are shown in the bottom row of Fig. 3b. For the first two states, there is excellent agreement between experimental and simulated maps. The energy difference between the third ($p = 2, q = 0$) and fourth ($p = 5/3, q = 2/3$) lowest energy states of a particle in a triangular box is small. Consequently, both states contribute to the experimentally observed contrast at $V = 0.17$ V.

## 3.2 Corral size and on-site energy

We now consider how altering the size of a corral affects the energies of the lowest levels, *i.e.* the on-site energies. We first focus on rectangular corrals. Fig. 4a shows a series of rectangular quantum corrals that were constructed. Differential conductance spectra acquired at the centers of the corrals are shown in Fig. 4b. As the corral is reduced in size, the ground state shifts to higher energies. The second peak for S1 at higher energies corresponds to a $2s$-like orbital, *vide infra*. Note that peaks become progressively broader with increasing energy. We attribute this to two factors. First, the scattering potential of the CO molecules is finite (0.9 eV with respect to the onset of the surface state band when a radius of 0.3 nm is used). Hence, electrons with higher energy effectively experience a lower barrier height. Secondly, the number of CO molecules per unit area is larger for smaller corals, resulting in an increased coupling between surface and bulk states [44].

To rationalize the experimental observations, we model our system using a particle-in-a-box model with *finite* potential barriers of height $V_0 = 0.9eV$ [36]. For a 2-dimensional rectangular box with finite barriers, the energies are given by

$$E = V_0 - \frac{2\hbar^2}{m^*}\left(\frac{u_{n_x}^2}{L_x^2} + \frac{u_{n_y}^2}{L_y^2}\right), \tag{6}$$

where $m^* = 0.42m_e$, the effective mass of the Cu(111) surface state electrons, and $L_x$ and $L_y$ correspond to the length of the box in the $x$ and $y$ direction, respectively [36]. The variables $u_{n_x}$ and $u_{n_y}$ take the role of quantum numbers. Their values are the solutions to the following set of three equations (where $i$ denotes the $x$ or $y$ direction of the rectangular box)

$$u_i = \sqrt{u_{0_i}^2 - v_i^2}, \tag{7}$$

$$u_i = v_i \tan(v_i), \tag{8}$$

$$u_i = -v_i \cot(v_i), \tag{9}$$

with $u_i = \frac{\sqrt{2m^*(V_0-E)}L_i}{2\hbar}$, $u_{0_i} = \frac{\sqrt{2m^*V_0}L_i}{2\hbar}$, and $v_i = \frac{\sqrt{2m^*E}L_i}{2\hbar}$. No analytical solutions exist for these equations and one has to rely on graphical or numerical methods [36]. The solutions are given by the values of $u_i$ where function (7) intersects function (8) or (9), and are denoted

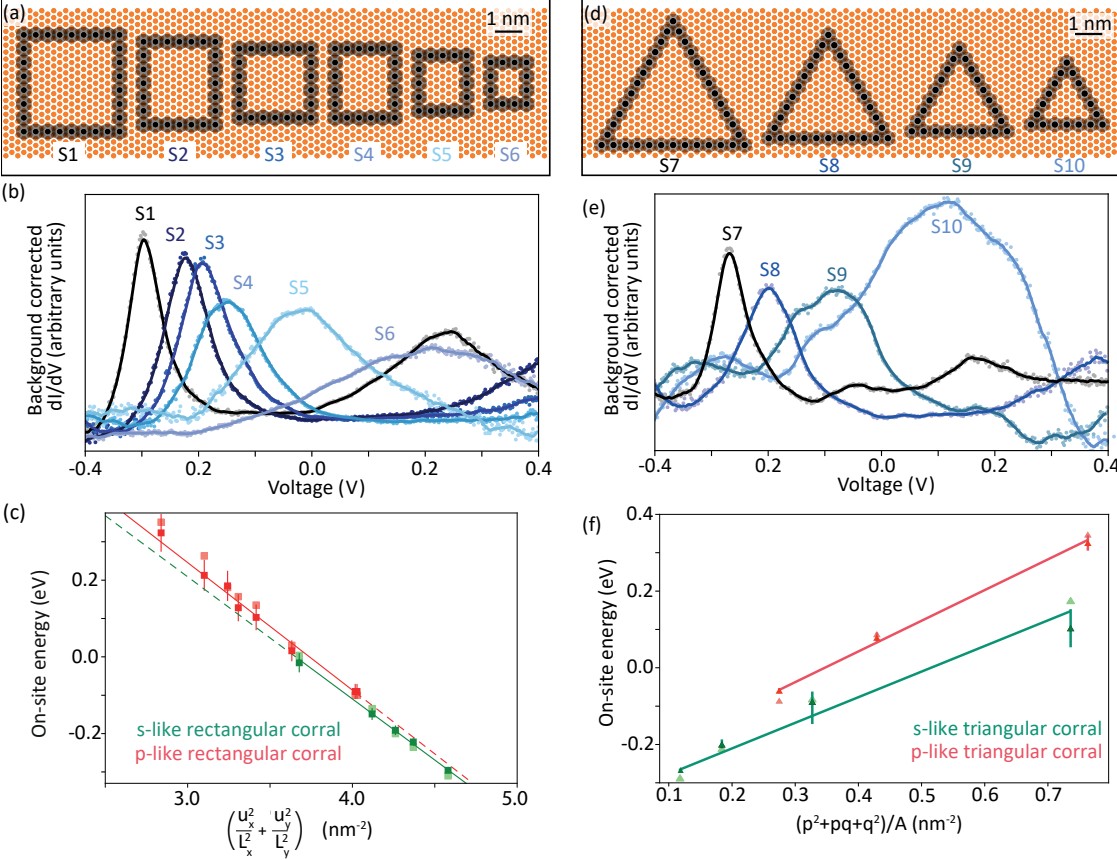

Figure 4: (a) Geometries of the rectangular corrals investigated. (b) $\frac{dI}{dV}$ spectra taken at the centers of the rectangular corrals shown in (a). (c) On-site energy as a function of $u_{n_x}^2/L_x^2 + u_{n_y}^2/L_y^2$. Green and red points represent experimental data for $s$- and $p$- like states respectively. Light red and light green points represent the energies calculated using a muffin tin model. Solid lines represent a linear fit to the experimental data. (d)-(f) Same as (a)-(c) but for triangular corrals.

$u_{n_i}$. For a given $V_0$, $L_i$ and effective mass, the values of $u_{n_i}$ are fixed, and can be thought of as analogous to the quantum number in the energy equation that describes a particle in a 2D rectangular box with infinite barriers. To calculate the values of $u_n$, we use $V_0 = 0.9$ eV [27, 29–31, 33] and $m^* = 0.42m_e$ [45, 46]. The values of $L_x$ and $L_y$ are determined by assuming that the dimensions of the boxes are defined by the edges of the CO molecules which have a diameter of 0.6 nm [27, 29–31, 33].

Figure 4c shows a plot of the on-site energy versus $u_{n_x}^2/L_x^2 + u_{n_y}^2/L_y^2$ for the lowest and second lowest states. Dark (light) green and red (light red) correspond to experimental (muffin tin) data of $s$-and $p$-like states, respectively. The experimental energies were determined by fitting Gaussian curves to each peak and finding the centers. The muffin tin-derived energies were calculated with the aforementioned values for $V_0$, $m^*$ and CO diameter. For both states, the energy depends linearly on $u_{n_x}^2/L_x^2 + u_{n_y}^2/L_y^2$. From the gradient, we determine the effective electron masses to be $0.48 \pm 0.01 m_e$ and $0.46 \pm 0.01 m_e$ for the $s$-like and $p$-like states, respectively. These values are close to the effective electron mass of the unconfined surface state electrons.

A small offset is visible between the lines for the $s-$ and $p$-like data, which we attribute to the fact that the confining potential is effectively lower for higher energy states.

We applied a similar procedure to triangular corrals. Figure 4d shows the geometry of several triangular corrals that were realized, and Fig. 4e shows spectra acquired at the centres. The states of triangular corrals shift to higher energies the smaller the corral becomes. The data can be rationalized using a particle-in-a-box model using infinite barriers (analytical solutions for triangular corrals with finite barriers have not been reported). The energy eigenvalues of a particle in an equilateral triangular box are given by

$$E_{p,q} = \frac{h^2}{2\sqrt{3}m^*A}(p^2 + pq + q^2),$$ (10)

where $p$, $q$ are the quantum numbers, $h$ is Planck's constant, $m^*$ is the effective electron mass and $A$ is the area of the triangle [35,47,48]. As shown in Fig. 4f, the experimentally determined on-site energy depends linearly on the inverse surface area, in agreement with equation (10).

## 3.3  Coupling Corrals

We now turn to coupled quantum corrals and show how tight-binding parameters can be extracted from experimental data. After a dimer is constructed (example shown in Fig. 5a), $\frac{dI}{dV}$ spectra are acquired on two positions. We do this to make use of the different spatial localization of the $E_+$ and $E_-$ states. Specifically, the anti-bonding $E_-$ state has a node between the two corrals (the position denoted by an orange dot in the inset of Fig. 5a). Only the bonding $E_+$ state appears in the differential conductance spectrum taken at that site and we can fit the spectrum with a single Gaussian. Conversely, the anti-bonding $E_-$ state has higher intensity at the outer regions of the dimer (red dot in Fig. 5a).

Differential conductance maps were acquired at approximately the energies of the centers of each of the two peaks. The state at lower energy is delocalized over the entire structure, whereas the state at higher energy has a node between the two corrals. This is reminiscent of bonding and anti-bonding molecular orbitals, respectively.

Next, a trimer is constructed from the same-sized units as the dimer. To determine the experimental values of $E_1$, $E_2$ and $E_3$, we again exploit the different spatial distributions of these three states. Muffin tin calculations show that the intensity of the $E_2$ state is very low at the center corral. Hence, the two peaks in the differential conductance spectrum taken at this position (gray curve in Fig. 5d) can be assigned to $E_1$ and $E_3$, respectively. The obtained energies can then be used in the fitting procedure of the spectrum acquired at a corral at the end of the trimer (red curve in Fig. 5c and d). Taking these values and solving equations 1 to 5 results in the tight binding parameters listed in Table I.

Table 1: Tight binding parameters extracted from Fig. 5.

| Parameter | Value |
|:---:|:---:|
| $\epsilon_1$ | $-0.22 \pm 0.02$ eV |
| $\epsilon_2$ | $-0.23 \pm 0.01$ eV |
| $s$ | $0.5 \pm 0.3$ |
| $t_1$ | $-0.14 \pm 0.06$ eV |
| $t_2$ | $-0.02 \pm 0.03$ eV |

The on-site energy of the individual corral of this size is $-0.19 \pm 0.02$ eV, see Fig. 4c. We find an on-site energy of $-0.22 \pm 0.02$ eV and $-0.23 \pm 0.01$ eV for the sites in the dimer and central site in the trimer, respectively. This lowering of the on-site energy can be understood from the increased area that is available due to the removal of the CO molecules to couple the sites. Two CO molecules have been removed from the barrier, $i.e.$ an additional area of $2 \times \pi(0.3)^2 = 0.56$ nm$^2$ is available for the electrons. The magnitude of the overlap integral,

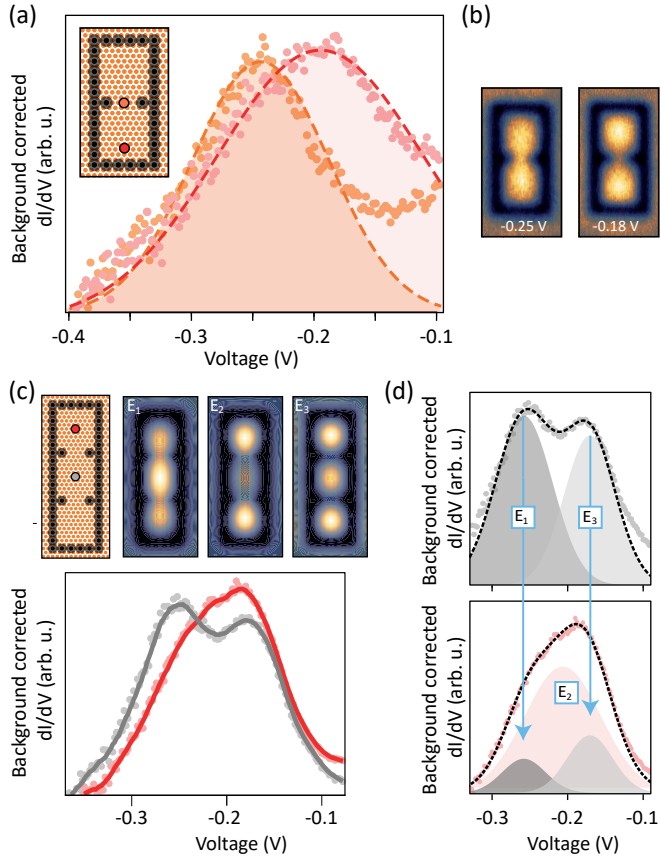

Figure 5: (a) $\frac{dI}{dV}$ spectra on two different positions of a dimer. Locations indicated by dots in the inset. (b) Differential conductance maps were acquired at the approximate energies where the maxima of the peaks lie. (c) Bottom: $\frac{dI}{dV}$ spectra acquired at the positions shown in corresponding colors in the top left diagram. LDOS maps (from muffin tin calculations) at $E_1$ = -0.26 eV, $E_2$ = -0.22 eV, $E_3$ = -0.18 eV, respectively. (d) Gaussian fitting procedure applied to the same two spectra to find the energies of interest. The shaded regions in each plot represent the individual Gaussians, which when summed, lead to the curves represented by dashed lines. The centers of the Gaussians correspond to $E_1$, $E_2$ and $E_3$ (labeled).

$s$, is significant and therefore must be included in tight binding parameters to yield accurate answers.

The same experiments and simulations were performed for coupling triangular corrals.

## 3.4 Tuning parameters

We now systematically investigate how the tight binding parameters depend on changing the gap width between corrals for both $s$- and $p$-like states. For this, we created dimers out of rectangular quantum corrals with dimensions $6\sqrt{3}a \times 10a$ (same as in the previous section) and $8\sqrt{3}a \times 14a$. (Note that to calculate the area from these dimensions, the area that the CO molecules occupy must be subtracted). First, two corrals of equal size were constructed directly next to each other with the barrier fully closed; that is to say that the same barrier configuration that separates the two corrals separates the corrals from their surroundings. Fig. 6a shows the schematic of a lone corral with dimension $8\sqrt{3}a \times 14a$, and the dimer with a full wall of CO molecules separating the corrals. The second column shows spectra taken at the positions marked in the designs. The peaks associated with the $s$-type orbitals occur at the

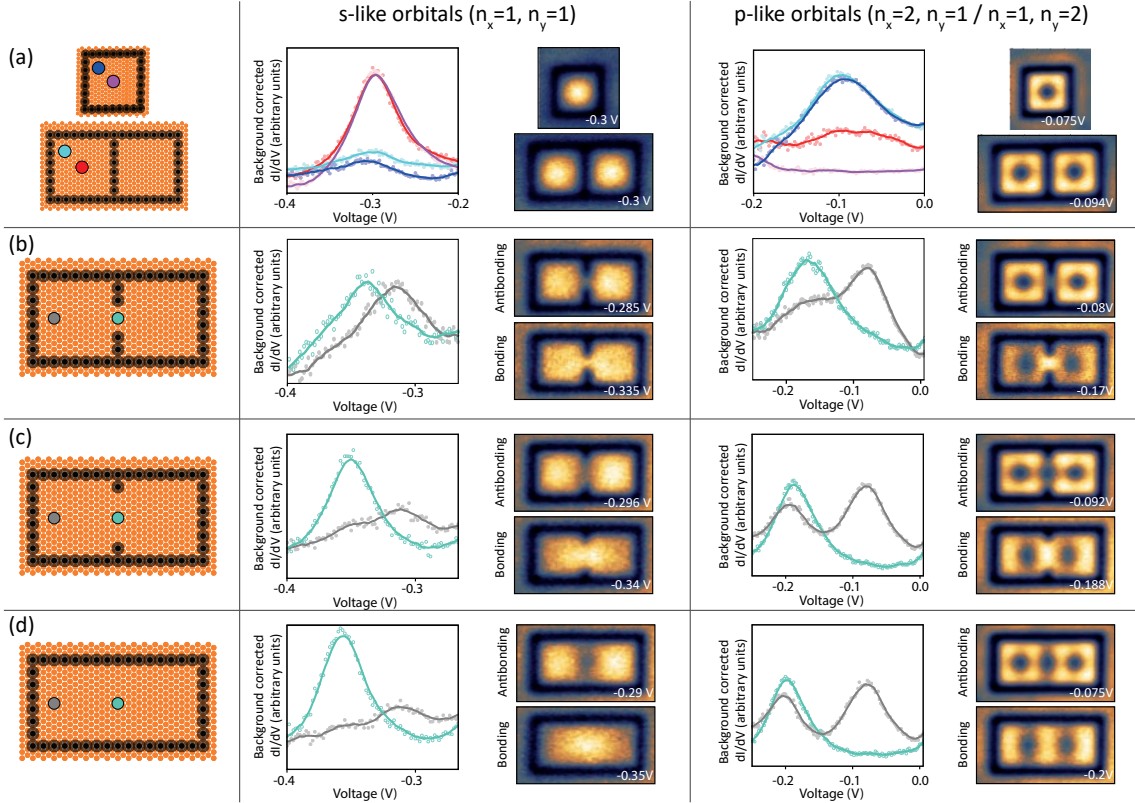

Figure 6: (a) From left to right, first the geometry of the coupled corral is shown. The second column demonstrates coupling of $s$-like orbitals: $\frac{dI}{dV}$ spectra acquired at the positions indicated by the color code and corresponding $\frac{dI}{dV}$ maps taken at the indicated energies. The third column focuses on coupling of $p$-like orbitals.

same energy for the two systems, indicating that there is virtually no coupling between the corrals in the dimer with this barrier configuration (the hopping parameter is zero). The same observation is made for the $p$-type states (right hand side of the figure). This is significant because it has been assumed that coupling of electronic sites to the surrounding 2DEG plays a large role in broadening [31]. Our experiments provide an upper-limit to the coupling strength across a 'full barrier': any potential splitting of the bonding and antibonding states is smaller than the energy resolution of our experiments. A muffin-tin calculation using small broadening finds a peak splitting of 11 meV (suggesting an upper limit of the coupling strength of 6 meV).

Next, CO molecules are removed from the center of the barrier, see Fig. 6b-d. As described before, $\frac{dI}{dV}$ spectra were acquired at the barrier between the corrals, and near the outer edge. By fitting Gaussian curves and finding their centers, we determine the energy level spacing between the bonding and anti-bonding states. Differential conductance maps were taken to verify the resemblance of these states to bonding and antibonding orbitals. The difference in energy between the two states increases with increasing gap width in the CO barrier between the two corrals. Furthermore, the states shift down in energy due to the effectively larger area that the electrons can occupy.

The most natural interpretation of the experimental data for the system without barrier, Fig. 6d, is to use a particle-in-a-rectangular-box model. In this picture, the lower energy state corresponds to the ground state with quantum numbers $n_x = 1$ and $n_y = 1$. The second state is the $n_x = 2$, $n_y = 1$ state, etc. However, it is also possible to interpret the results in the framework of two coupled quantum corrals. The lowest energy state of the rectangle can be thought of as the bonding combination of $s$-like orbitals of the two quantum corrals. Similarly,

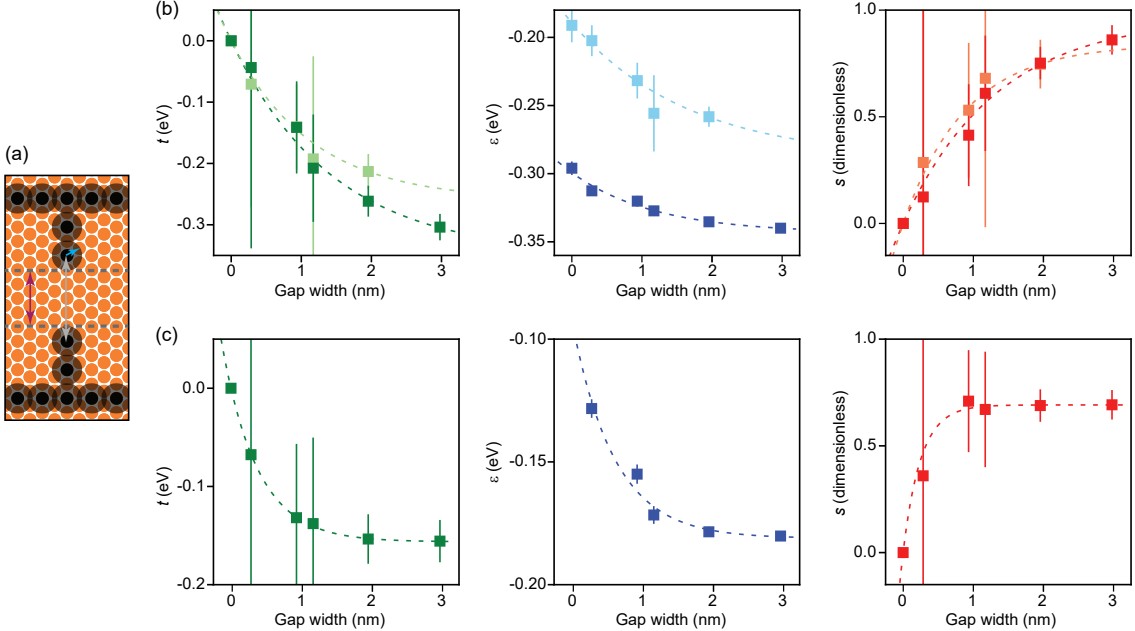

Figure 7: (a) The size of the gap in the barrier between the two corrals (red arrow) is the distance between the closest CO molecules in the barrier (gray arrow), minus two times the apparent size of the CO molecules (gray circle, blue arrow, 0.3 nm). (b) From left to right: gap dependence of the hopping parameter, the on-site energy and overlap for $s$-like states, respectively. Dark and right colors represent data from rectangular corrals with sizes $8\sqrt{3}a \times 14a$ and $6\sqrt{3}a \times 10a$, respectively. (c) Same as (b) but now for $p$-like states.

the second lowest state would be the anti-bonding combination.

The bonding combination of the $p_x$-like states, where $x$ is the horizontal direction, shows vertical nodal lines at the centers of the individual corrals and enhanced intensity in the barrier region between the corrals (see right hand side of Fig. 6). The nodal line pattern of the map at higher energy can be rationalized by assuming that both the anti-bonding $p_x$-like state as well as the $p_y$-like state contribute to the contrast. The energy difference between $p$-like bonding and antibonding states is larger than for the $s$-like states.

Similar experiments were performed for coupled $6\sqrt{3}a \times 10a$ dimers (data not shown). From the available data on both corral sizes, tight binding parameters for coupling of both $s$-like and $p$-like states were derived. The results are shown in Fig. 7. The size of the gap in the barrier between the corrals is defined as the distance between the closest CO molecules of the barrier, minus two times the apparent radius of the CO molecules (0.3 nm, see Fig. 7a). For both $s$- and $p$-like states and for both corral sizes, the data points for the hopping parameter ($t$), the on-site energy ($\epsilon$), and the orbital overlap ($s$) can be fitted with an exponential function (dotted lines). By tuning the gap width, the hopping parameter can be varied between 0 eV and $\sim -0.3$ eV and $\sim -0.16$ eV for $s$- and $p$-like states, respectively. We find that the on-site energy depends on the width of the gap in the barrier. The parameters depend more sensitively on gap width for the smaller corral. This can be rationalized from the additional area that becomes available to the confined electrons upon removing CO molecules (the relative increase in available area is larger for the smaller corral). Finally, the magnitude of the orbital overlap increases with gap width. Note that for unconfined electrons (infinite gap width) the overlap should be one. Fig. 7b suggests that at least up to a gap width of $\approx 1.5$ nm, the hopping parameter and overlap are similar for the two different corral sizes.

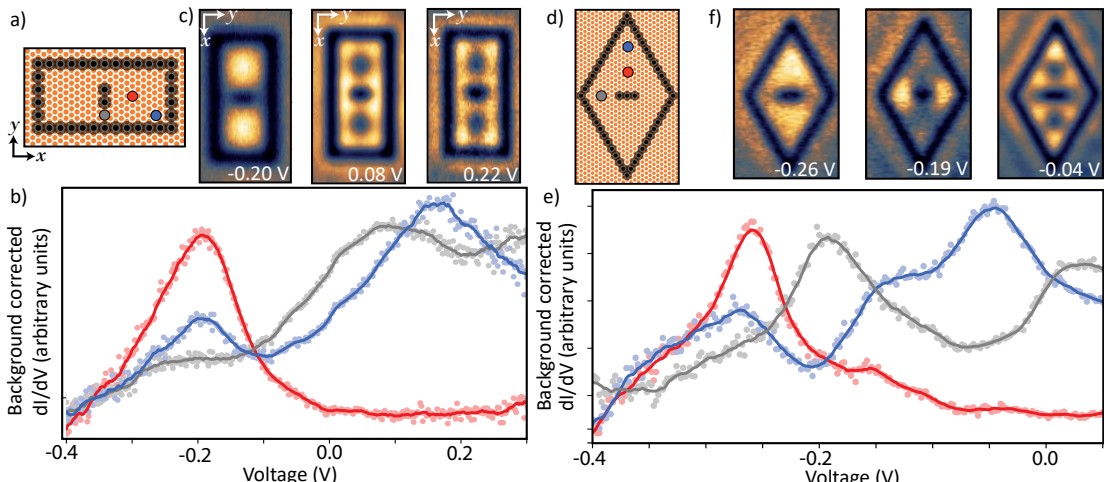

Figure 8: (a) Coupled rectangular corrals with a barrier that largely inhibits the coupling of $s$ and $p_x$-type states, while coupling of $p_y$-like states is clearly observed. $x$ and $y$ directions are specified in the figure. (b) Differential conductance spectra taken at the positions indicated in (a). (c) Differential conductance maps taken at the indicated energies. (d-f) same as (a-c) but for coupled triangular corrals.

## 3.5 State selective coupling

Since CO molecules can be removed selectively, it becomes possible to create geometries that allow coupling of $p$-type states only. Consider the geometries of coupled rectangular and triangular corrals shown in Fig. 8a and d. The amplitude of $s$-type wave functions is small at the position of the gaps in the barrier. Hence, coupling of $s$-type states should be small. In contrast, $p$-type states have significant amplitude at these positions and consequently these states should couple strongly. We first focus on the rectangular corrals. Fig. 8b shows differential conductance spectra taken at the positions indicated in Fig. 8a. A total of three peaks are observed. The amplitude of each peak differs from position to position. The peak at lowest energy corresponds to the ground state, *i.e.* it involves $s$-type states. At the energies corresponding to the $s$-type states, we only observe one peak, indicating that these states do not couple (coupling strength below the detection limit of our experiment). In contrast, the spectrum of the barrier region (gray) features a peak around 90 mV, whereas the spectrum taken at the corner of the corral (blue) has a peak at 170 mV. The corresponding differential conductance maps, Fig. 8c, reveal that the spatial extent of these states can be understood by considering coupling of $p_y$-type states. For the triangular corral, similar observations are made. This confirms the idea that artificial lattices allow coupling between sites by one type of state only [49]. Note that this provides a degree of freedom that is not available in real materials.

## 3.6 Coupling corrals of different sizes

Finally, we investigate the coupling of two corrals of different sizes, *i.e.* with different on-site energies for the $s$- and $p$-like states. Fig. 9a shows the arrangement of such a polar dimer, with the barrier between corrals fully removed to maximize coupling. The $\frac{dI}{dV}$ spectra show the typical peaks associated with bonding and antibonding states. The corresponding differential conductance maps reveal that the lower (higher) energy state of the dimer is primarily localized on the larger (smaller) corral, see Fig. 9b and c. This is in agreement with a tight binding model of a dimer with constituents with different on-site energy.

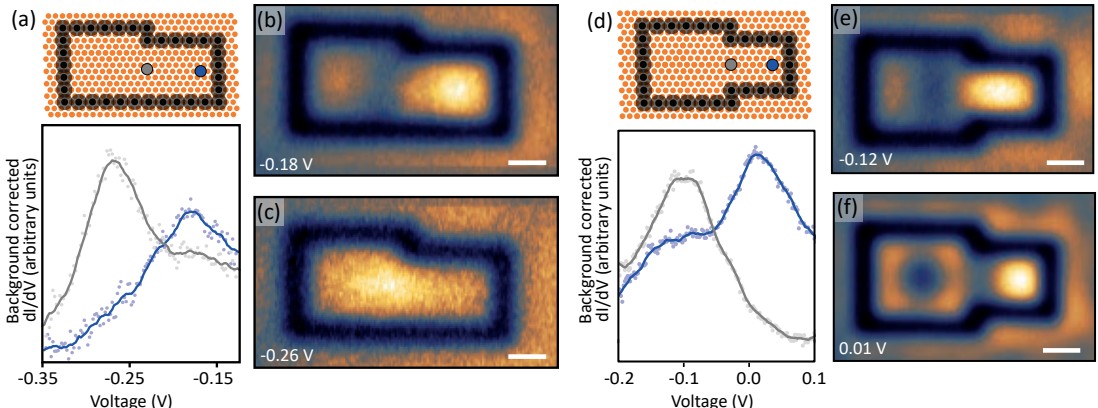

Figure 9: (a) Schematic structure of anisometric dimer consisting of a $6\sqrt{3}a \times 12a$ corral coupled to a $5\sqrt{3}a \times 10a$ corral. Differential conductance spectra acquired on positions highlighted in the inset. (b),(c) differential conductance maps of the two states observed in the $\frac{dI}{dV}$ spectra (energies indicated in the Figure). (d)-(f) same as (a-c), but now for a $6\sqrt{3}a \times 12a$ corral coupled to a $4\sqrt{3}a \times 8a$ corral. Inset scale bar (white) represents a length of 1 nm

In general, electronic states couple if they spatially overlap and if they have a similar energy. Hence, if the sizes of the two corrals differ sufficiently, it is possible to couple the *s*-like state of a smaller corral with a *p*-like state of a larger corral. We therefore created a dimer consisting of a $6\sqrt{3}a \times 12a$ to a $4\sqrt{3}a \times 8a$ corral, see Fig. 9d. The $\frac{dI}{dV}$ spectra reveal two states with different spatial localization. The corresponding differential conductance maps show that the lower energy *s*-like state of the smaller corral couples with a *p*-like state of the larger corral. Similarly, the higher energy state can be thought of as an antibonding combination between *s*- and *p*-like states (note the nodal line at interfaces between the two corrals).

# 4   Conclusion

To conclude, we have studied the coupling of rectangular and triangular quantum corrals into dimer and trimer structures. These shapes were chosen as they can be used as building blocks of artificial lattices. The electronic structure of the coupled corrals can be understood using a tight binding model also used for the coupling of atoms to molecules. Importantly, we investigated the available tight binding parameter space accessible with the CO/Cu(111) platform, and showed how these parameters depend on the configuration of the coupled quantum corrals.

We first verified that the particle in a box model provides a good qualitative description of the electronic structure of rectangular and triangular quantum corrals. We determined the onsite energies of *s*- and *p*- like states of different sized corrals to confirm the relationship between on-site energy of the corral and box size. From this, we determined the effective masses of electrons in rectangular corrals to be on the order of 0.48 $m_e$ and 0.46 $m_e$ for *s*- and *p*- like states respectively. These values are close to the value for unconfined Cu(111) surface state electrons (0.42 $m_e$). In the triangular case, we used a model that assumed infinite barriers, preventing us from determining a reliable value for the effective mass.

We outlined a method to extract tight binding parameters (nearest and next nearest neighbor hopping parameters, overlap and on-site energy) by constructing dimers and trimers of corrals. By removing CO molecules from the barrier between corrals, exponential relationships were found between the tight binding parameters and the size of the gap in the barrier

between the corrals. The hopping integral can be tuned between 0 and -0.3 eV and -0.16 eV for $s$- and $p$-like states, respectively, by tuning the configuration of CO molecules in the barrier. In most cases, the overlap is not negligible and this term should be taken into account when modelling artificial molecules, and lattices. Finally, we showed that in these coupled quantum corrals, one can control which states couple. For example, by appropriate placement of CO molecules coupling of $s-$ and $p_x$-like states can be inhibited, while allowing coupling of $p_y$-like states. Furthermore, it is possible to couple $s$- and $p$-like states.

The results presented here are useful for future work on artificial lattices made using CO on Cu(111). A hypothetical lattice with certain desired coupling strengths and on-site energies can be designed by estimating the required unit size and barrier gap width from the trends reported here.

# Acknowledgements

We thank Daniel Vanmaekelbergh and Jette van den Broeke for useful discussions.

**Funding information** We kindly acknowledge funding from the Dutch Research Council (NWO) via grant 16PR3245.

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
