# Peer review of "Coupling quantum corrals to form artificial molecules"

_SciPost Physics, doi:SciPost Phys. 9, 085 (2020)_

## Round 1 · Referee Report · Robert Drost (Referee 1) · 2020-10-19

Strengths

Freeny and co-workers present a thorough documentation of the electronic properties of quantum corrals formed by confining the electrons in the surface state of a Cu(111) surface by targeted assemblies of CO molecules. The authors investigate the influence of the shape and size of the corral on the energy level positions and broadening and show how tight-binding parameters describing coupled corrals can be extracted from the experiment. They continue to demonstrate how these parameters may be tuned through structural changes to the corrals and discuss strategies for engineering non-standard couplings.

Report

The work is carefully executed and the data presented of excellent quality. The most important contribution from the paper is a comprehensive mapping of the available parameter space for studies of artificial lattices constructed in the CO/Cu(111) template. I recommend to publish the manuscript with minor revisions as noted below.

Requested changes

1.- Concerning Fig. 2, the centre corral of the trimer has a slightly different area and confinement potential from those at the ends and therefor does not share the same wave function with the end sites. The overlap integral s in Eqs. (1) and (2) and that in Eqs. (3) through (5) are not identical. While I think that the approximation to treat them as such is justified, it would be good to briefly acknowledge this in the text.

2.- Fig. 3 uses the label 'Corrected dI/dV' while all other figures with conductance spectra read 'background corrected dI/dV'. Are both referring to the same procedure described in section 2 of the manuscript?

3.- What are the faded dots that appear in the plots of conductance spectra? I assume that the lines represent the averages mentioned in section 2. Are the dots raw data? It would be good to have a brief explanation if both are to be shown.

4.- Concerning Fig.3, I believe that there must have been a mix-up with the marker and curve colours. The red and green curves are nearly identical, even though they are seemingly measured at different locations. The red and blue curve are qualitatively different even though they have been acquired at symmetrically (nearly) identical points of the corral. The blue curve is the only data set showing a signature of the (2, 2)-state even though it has a nodal plane at the indicated location. Please double-check the assignments.

5.- Concerning the relationship between the corral size and on-site energy discussed in Fig. 4, I estimate the slope of the graph in Fig. 4c to be ca. -0.33 eVnm2 and that in Fig. 4f ca. 0.66 eVnm2. There are two issues here: First, why is the slope in Fig. 4c negative? I get m0.46me from my estimate. The value is the same as that reported in the paper, but what of the sign? Second, the effective mass extracted from Fig. 4f comes to ca. 1.3me for me. This is somewhat surprising. Why should m depend on the shape of the confining potential? A brief discussion would be good.

6.- Eq. (7) is a little difficult to follow. Under what conditions of the upper and lower expression after the curly braces in the equation hold? What is V0 used in the definitions of u0 and vi? It seems that this is the height of the scattering potential discussed earlier, but there is no explicit assignment. The quantity ui is defined following Eq. (7), but does not seem to appear in it, nor in u0 or vi defined before.

7.- On page 11, end of the first paragraph, the authors write: 'The results reported here suggest that a full barrier would result in negligible coupling betwen states in a lattice and the surrounding 2DEG.' I do not believe that the data presented support the claim with this level of generality. Its is clear from Fig. 4 that the broadening of the corral states is a function of the on-site energy and/or the corral area. It is difficult to imagine another mechanism than life time broadening from particle loss to the 2DEG and bulk states.

8.- Concerning Fig. 7, I would consider using the number of missing CO molecules as the x-coordinate of the plots in panels b and c. It seems a much more intuitive length scale given to me.

  • validity: top
  • significance: good
  • originality: good
  • clarity: top
  • formatting: excellent
  • grammar: excellent

Author:  Ingmar Swart  on 2020-11-02  [id 1023]

(in reply to Report 1 by Robert Drost on 2020-10-19)
Category:
answer to question

We are grateful for the positive review that Robert Droste has submitted in response to our manuscript. We would like to thank him in particular for the time and effort he gave for his very thorough and insightful evaluation. This helps us to give clearer explanations and a fuller picture of our work. In the attached file, we reply in detail to all comments.

Attachment:

Robert_droste_response_final.pdf

---

## Editorial Decision

published